# Cost-Effective Vibration Analysis through Data-Backed Pipeline Optimisation

**DOI:** 10.3390/s21196678

**Published:** 2021-10-08

**Authors:** Artur Sokolovsky, David Hare, Jorn Mehnen

**Affiliations:** 1Department of Design, Manufacturing and Engineering Management, University of Strathclyde, Glasgow G1 1XJ, UK; jorn.mehnen@strath.ac.uk; 2Bosch Rexroth Ltd., Saint Neots PE19 2ES, UK; david.hare@boschrexroth.co.uk

**Keywords:** signal processing optimisation, IoT, MAFAULDA, vibration analysis, failure mode classification, machine learning, effect sizes, hypothesis testing

## Abstract

Vibration analysis is an active area of research, aimed, among other targets, at an accurate classification of machinery failure modes. The analysis often leads to complex and convoluted signal processing pipeline designs, which are computationally demanding and often cannot be deployed in IoT devices. In the current work, we address this issue by proposing a data-driven methodology that allows optimising and justifying the complexity of the signal processing pipelines. Additionally, aiming to make IoT vibration analysis systems more cost- and computationally efficient, on the example of MAFAULDA vibration dataset, we assess the changes in the failure classification performance at low sampling rates as well as short observation time windows. We find out that a decrease of the sampling rate from 50 kHz to 1 kHz leads to a statistically significant classification performance drop. A statistically significant decrease is also observed for the 0.1 s time window compared to the 5 s one. However, the effect sizes are small to medium, suggesting that in certain settings lower sampling rates and shorter observation windows might be worth using, consequently making the use of the more cost-efficient sensors feasible. The proposed optimisation approach, as well as the statistically supported findings of the study, allow for an efficient design of IoT vibration analysis systems, both in terms of complexity and costs, bringing us one step closer to the widely accessible IoT/Edge-based vibration analysis.

## 1. Introduction

Vibration analysis is a hot research topic that is interesting for both researchers and industry practitioners. Successful vibration analysis allows performing machinery condition monitoring, failure mode analysis, and even predictive maintenance. It is especially interesting in the context of the development of Internet of Things systems (IoT), allowing remote system diagnostics. However, when developing a practical and scalable vibration analysis IoT solution, one faces certain challenges. Namely, industrial-grade accelerometers can be expensive, making their use in the IoT setting limited. Another challenge is the system-specific data properties and failure modes which make generalising the findings across systems non-trivial. Finally, there is a general lack of public datasets in the field due to the purposeful induction of specific failure modes in a realistic way being a challenging and resource-demanding task. Due to the limited data availability, there is a demand for a methodology for a data-efficient signal processing pipeline optimisation. It is expected to require little or no extra data for optimisation. Moreover, it should be robust to overfitting and reasonably fast to evaluate. To the best of our knowledge, no uniform methodology has been proposed so far addressing all these issues.

The current body of knowledge proposes multiple ways of analysing vibrations and detecting faulty system states [1,2,3,4,5]. However, many studies are short on demonstrating the optimality of the proposed methods as well as justifying the introduced complexity [2,3,5]. Contributions of every pre-processing step to the final result are often not clear. There are also deep learning-based approaches leading to high classification performance, which are proposed as alternatives to the signal processing methods [4]. They often take the raw data as the model input applying no prior signal processing methods. We believe that it would be beneficial to systematically assess any of these design decisions.

In order to provide the reference classifier performance for the current study, as well as familiarise the reader with the recent works in the field, we discuss three studies, where the authors use the same performance metrics and classes (not considering bearing failure sub-types) as in the current work. First, in the work of Alzghoul et al. [6], the authors propose a set of features based on DFT (Discrete Fourier Transform), entropy, kurtosis and mean. They account for the dataset class imbalances using SMOTE (Synthetic Minority Over-sampling Technique). Additionally, they perform Relief feature selection. For classifying entries, the authors make use of an MLP (Multilayer Perceptron). Finally, the study reports an accuracy of the classifier of 97.1%. The purpose of the study is to investigate the benefits of using SMOTE and Relief methods together.

In the second study by Ali et al. [7], the authors intentionally use a shallow set of features together with SMOTE for class balancing and an MLP classifier reporting accuracy of 96.2%. The purpose of the study is to demonstrate the benefits of using SMOTE.

Finally, in the work by Marins et al. [8], the authors use a principally different approach to feature extraction—SBM (Similarity-Based Modelling). They investigate the effectiveness of this approach as a standalone method as well as the source of features for a Random Forest classifier. The reported accuracy on MAFAULDA dataset is 98.5%. As one can see, each of the studies focuses on different aspects of the classification pipeline, as well as uses different sets of features and classifiers. Moreover, cross-validation strategies vary between the studies. The mentioned degrees of freedom make it rather hard to compare the performances across studies. While the latencies are not reported in the reference papers, based on the provided implementation details, we can assume that the proposed approaches allow classification at sub-second evaluation times on single-board computers, like Raspberry Pi, making them potentially suitable for IoT purposes.

In a recent paper, Rauber et al. [9] propose a methodology for machine learning pipeline validation aimed at improving the reliability of the obtained performance. In the current work, we address a related aspect of the vibration analysis—the in-depth assessment of the signal processing pipeline components which allows a data-driven justification of the pipeline complexity. We do so by first optimising individual components of the signal processing stack and then searching for the optimal sequence of the components. This is done using statistical goodness of fit measures. Finally, we demonstrate how machine learning methods can be used in interplay with the proposed approach, getting the best of the two worlds, machine learning and signal processing.

Using the proposed methodology, we assess the feasibility of hardware cost optimisation in the industrial IoT setting. When considering the cost optimisation, we take into account not only the feasibility of high-performance pipeline design but also the computation limitations of the IoT devices. For instance, we do not expect IoT devices to have a GPU processor available, and we also make assumptions about the available CPU frequencies and RAM. The particular latency requirements depend a lot on the setting and should be adjusted on a case-by-case basis. Taking these limitations into account, we measure latencies of the optimised pipelines and discuss to what extent they are suitable for the failure classification of low-mid rotation frequency machinery in industrial settings. Among the reported configurations, there are some suitable for the sub-second operation latencies.

The contributions of the paper are the following:We propose a systematic approach to optimisation of the signal processing pipelines on the example of the MAFAULDA dataset [10];We assess whether lower sampling rates (1 kHz) used by cost-effective (and lower sampling rate) accelerometers lead to a statistically significant failure classification performance drop in comparison to the original 50 kHz sampling rate;Aiming to optimise the latency of the systems, we also investigate whether sub-second observation time windows lead to a statistically significant failure classification performance drop in comparison to the 5 s observation time windows;Lastly, we report the failure classification performance results for logistic regression and CatBoost [11] classifiers, as examples of the simple and robust, and cutting edge boosting trees estimators, respectively. We compare the performance to the existing studies and allow to assess in an exploratory way whether simpler and computationally faster classifiers are suitable for the domain.

The rest of the paper is structured as follows: in Section 2, we describe the dataset and detail the methodology of the study; in Section 3, we report the results of the experiments; then, we discuss the results in Section 4; finally, we conclude our work in Section 5.

## 2. Materials and Methods

The section is structured as follows. We first formulate the aim of the study and then describe the dataset, label design, sampling strategies, and the feature space. Then we list the signal processing methods and the way its hyperparameters and the order are optimised. After the signal processing is done, we apply the machine learning pipeline, including the feature space optimisation, hyperparameter tuning, and model performance evaluation. Finally, we answer the research questions by running statistical tests and computing effect sizes.

### 2.1. Aims

In the current study, we introduce the pipeline for systematic optimisation of the signal processing methods and their integration with the machine learning stack. Using the proposed pipeline, we answer two research questions by formulating the null and alternative hypotheses and then testing the null hypothesis for rejection by running statistical tests. The limitations and generalisation of the findings are discussed in Section 4.

**RQ1:** How does the data sampling rate affect the failure classification performance? Answering this research question would allow field practitioners to make informed decisions about the choice of the accelerometers and manage the budget more efficiently. We propose the following null and alternative hypotheses:
H${}_{01}$: 50 kHz sampled data allow the same or worse classification performance than the 1 kHz sampled data.H${}_{11}$: 50 kHz sampled data allow significantly better classification performance than the 1 kHz sampled data.

The lower sampling rate is chosen based on the sampling rates of the cost-effective devices available on the market.

To assess the latency limitations of the vibration analysis pipelines, we propose the second research question.

**RQ2**: How does the time window length affect the failure classification performance? We formulate the hypotheses as follows: H${}_{02}$: The failure mode classification performance on the data collected over 5 s is the same or worse than for the 0.1 s windows.H${}_{12}$: The failure mode classification performance on the data collected over 5 s is significantly better than for the 0.1 s windows.

The reference window size (5 s) is the longest available in the dataset. It is not feasible to study a gradual decrease of the observation window size preserving the optimisation space size. Hence, we set its size to an arbitrarily small value, where detection of the system frequencies is still theoretically possible.

These research questions are valid for the considered system rotation frequencies. The current dataset is considered as a statistical sample of convenience, while the statistical population would involve data of comparable operation frequencies and data quality. The model performance is defined as the mean absolute error of the model output on per-class probabilities [12].

### 2.2. Dataset

In the current study, we use the MAFAULDA (MAchinery FAULt DAtabase) vibration dataset, which provides vibration data sampled from two 3-axis accelerometers with a 50 kHz frequency [10]. The data is recorded from a physical system with the capabilities of modelling multiple failure modes, running at frequencies from 11.7 Hz to 60 Hz and operating on 8-ball bearings. The dataset is structured as independent 5 s time windows (250 k datum entries in each). The time windows are collected at different system rotation frequencies. This gives a detailed picture of how the system operates in different failure modes and a range of rotation frequencies.

#### 2.2.1. Label Design

There are 6 high-level system modes of operation: normal, horizontal, and vertical misalignment of the shaft; shaft imbalance; and overhang and underhang bearing failures: cage fault, outer race fault, and ball fault. Since this study is not aimed at claiming a state-of-the-art performance in the failure mode classification but rather at demonstrating the approach and answering the research questions, we use the 6 failure modes as the classification labels not distinguishing between different types of bearing failures, but only distinguishing the corresponding bearing.

#### 2.2.2. Sampling Strategies

To answer the research questions stated in Section 2.1 of the paper, we sample the raw accelerometer readings in three ways:1To mimic the real-world limitations of the cost-effective IoT vibration analysis solutions, we sample every 50th data point, representing the 1 kHz sampling frequency. The full available observation time window is used, leading to 5 k data points entries.2Aiming to study how the window length affects the performance, we obtain the first 5 k points with a 50 kHz sampling rate, corresponding to the 0.1 s observation time window.3First and second sampling strategies are compared to the full available time window of 5 s and the maximum available sampling rate of 50 kHz.

#### 2.2.3. Feature Space

The existing body of knowledge proposes various feature spaces for vibration analysis, including using raw vibration readings in deep learning models and designing domain-specific feature spaces. In the current work, we aim to make the findings of the study generalisable; hence, we use an open-source Python-based software package for time series feature extraction—TSFEL [13], version 0.1.4 (the latest available at the moment of conducting the experiments). We use its spectral-domain features. This feature space is used uniformly at all the stages of the study—in the signal processing and machine learning pipelines.

#### 2.2.4. Cross-Validation

When reporting the results, we split the data into training and test batches using a K-fold approach with 3 folds. The data are split into training and test 3 times so that the test parts of the folds reconstitute the whole dataset. This allows us to get test results for the whole dataset. We split the data into folds before any optimisations and preserve the splits across all the stages. The predictions are concatenated across the folds before computing the effect sizes and running the statistical tests. Machine learning optimisation is done on the training batch and at every step includes internal cross-validation policy with the 3-fold split. We withhold from running the nested cross-validation as this would significantly increase the computation times making the experiments infeasible.

### 2.3. Signal Processing Pipeline

Guided by the existing body of knowledge, we take the components of a signal processing pipeline proposed for lower frequency systems [2] and comprised of continuous wavelets [14], non-linear energy operator [15], CEEMD [16], and envelope transformation [17]. The study does not systematically optimise the hyperparameters and the order of the steps. In our design, we address this point by assessing the performance of the individual pipeline components and different component sequences. Since CEEMD is computationally very demanding, we use its simpler version—EMD [18].

EMD is an adaptive noise reduction approach that is commonly applied to non-linear and non-stationary time series. It breaks the non-stationary time series into intrinsic, finite components or Intrinsic Mode Functions (IMFs). The non-linear energy operator computes the energy of the signal in a non-linear fashion, allowing it to successfully capture the characteristics of the systems. Being non-linear, it allows spotting properties of the data missed by the Fourier transform. Wavelets are a widely used method in many fields, starting from data compression and ending with image processing. They are also commonly used for data approximation and denoising, where the unwanted frequencies are filtered out.

Since the whole process is computationally intensive, we make a design decision to exclude envelope transformation from the list of methods. We are aware that envelope analysis has been successfully used for bearing fault diagnosis [19]. While excluding it from the optimised methods might negatively impact the performance, the current study demonstrates a proof of concept of a very flexible approach, where the particular pool of signal processing methods is chosen based on the nature of data, available computation resources, and the limitations of the target IoT device.

Consequently, we end up with the signal processing pipeline comprising of 3 methods to optimise: wavelets, non-linear energy operator, and EMD. In the pipeline, we do not make any assumptions about the optimal sequence of the methods, and instead, we search for it by iterating over all possible variants. In the case of the larger component spaces, it becomes infeasible, and some constraints should be introduced. The high-level diagram of the steps is provided in Figure 1.

Since the detection of failures can be done by manually assessing the frequency domain of the vibration spectra and associating particular peaks with the failures, we assume that linear effects are strong in the data. Hence, we use a logistic regression model and Akaike’s Information Criterion (AIC) [20] to assess its goodness of fit for the signal processing optimisations. AIC allows us to compare across different signal processing pipeline configurations—both hyperparameters and the order of the methods. While one could use more advanced methods involving cross-validation, it would have been more computationally intensive and also required extra data. Considering the setting and the complexity of getting high-quality labelled data, we aim to minimise the use of extra data in the optimisation process.

#### 2.3.1. Optimisation

As the first step, we optimise the hyperparameters of the pipeline components. We discuss the weaknesses of this design decision in Section 4.3. Namely, there are two which require parameter optimisation: wavelets and EMD. Wavelets are optimised by the scale range and the type of wavelet and EMD by the number of intrinsic mode functions (IMFs).

After the hyperparameters are tweaked, there is a need to justify the order of the applied methods. To do so, we compute the goodness of fit on all the possible permutations of the methods, including the goodness of fit on the feature space without any signal processing applied. For both stages, we remove the 0-variance feature and standardise feature values before feeding into the logistic regression.

#### 2.3.2. Latency Tests

To ensure the feasibility of practical use of the considered signal processing methods as well as the whole pipeline, we measure the evaluation times of the signal processing and feature extraction components. After the hyperparameters and the component order are optimised, we run the latency tests. The test involves processing 100 data entries (observation windows) with further reporting of the mean and standard deviation of the evaluation time. Aiming to mimic the IoT setting, we cap the CPU frequency at 1.4 GHz and available RAM at 2 Gb. These resources are typically available on single-board computers like Raspberry Pi. Finally, we do not use parallelisation. We process data from two 3-axis accelerometers which sums up to 6 data streams processed sequentially.

### 2.4. Machine Learning Pipeline

After the signal processing pipeline is optimised and its complexity is backed by the data, machine learning methods are applied to the pre-processed data.

When performing the experiments, we want to cover the performance-oriented and latency-oriented scenarios. Hence, we classify the data using logistic regression and CatBoost classifiers [11].

#### 2.4.1. Feature Selection

While we do not perform any feature space optimisation at the point of signal processing, there is a need to do so if one aims to get good classification performance. We use recursive feature elimination with cross-validation (RFECV) with a decreasing step and class-weighted f1-score as the performance metric [21]. While there are less computationally intensive methods, RFECV is widely accepted across the ML community. It also aligns well with the data-backed approach of the current study.

#### 2.4.2. Hyperparameter Tuning

After the feature space is optimised, there is a need to adjust the hyperparameters of the models. In the logistic regression, we optimise the regularisation parameter, and for CatBoost—the maximum depth of the trees, regularisation, and the number of trees.

#### 2.4.3. Model Evaluation

To give a complete picture of the model performance, we report multiple classification performance metrics. Namely, we report f1-score, precision, recall—weighted by class, and accuracy. F1-score is known to work well with class-imbalanced datasets, and precision and recall are reported as its components [22]. Accuracy provides a fraction of correctly classified entries. We claim that accuracy is a valid metric to use in the considered case as the class imbalance in the dataset is moderate (reported in Table 1), and the majority class will not lead to the artificially high metric values.

#### 2.4.4. Latency Tests

Additionally, we assess the latency contribution from the considered machine learning models by measuring the test phase evaluation times. The evaluation times are measures per single classified entry over 100 iterations with mean and standard deviation reported. Here, we use the same hardware setup as in the signal processing latency measurements.

### 2.5. Statistical Evaluation

We perform the analysis on the absolute differences between the predicted class probabilities and the actual labels in a per-entry fashion. Hence, for each classified entry, we obtain a vector of positive values from 0 to 1, of the length equal to the number of classes. From there, we compute the mean absolute error (MAE) for the whole dataset. To formally answer the research questions, we perform hypothesis testing—this allows us to formally reject the null hypotheses. Moreover, we compute the effect sizes to quantify the strength of the effect. Moreover, by reporting the 0.95 Confidence Intervals (CIs), we generalise the effect sizes to the statistical population.

As an effect size measure, we use Hedge’s $g_{av}$. It computes the mean group difference corrected for paired entries [23]. We test the hypotheses using the Wilcoxon test [24], which is a non-parametric version of a t-test applied in cases of paired measurements and different variance between the studied groups.

Since we report multiple statistical tests on related data, it is necessary to account for the multiple comparisons. We do so by correcting the significance levels of the tests using Bonferroni corrections [25]. Since all the tests are performed on the related data, we treat all the tests of the study as a single experiment family. Hence, the corrected significance level is $\alpha_{corr}=\alpha /n$, where *n* is the number of tests. Corrections for multiple comparisons are also applied to the confidence intervals of the effect sizes.

## 3. Results

### 3.1. Dataset

The distribution of the failure modes by the number of collected entries is provided in Table 1. As it can be seen, around half of the entries are related to the bearing failures and the least number of entries are available for the normal machine state.

### 3.2. Signal Processing Pipeline

The optimisation of the signal processing methods is carried out for two sets of hyperparameters, depending on the number of data points (Table 2). Namely, for the 250 k data points, we shrink the number of hyperparameter values to stay in the feasible range of required computation power. The implications of this design decision are discussed in the limitations of the paper. Since the experiments are parallelisable across the data entries, we use a High Performance Computing cluster to run the experiments, taking around 20 k CPU hours in total. Each of the experiment configurations might take from hours to weeks of processing, depending on the available computation resources. Running the experiments for the current study, we use around 400 CPUs in parallel and all the experiments are completed within around 2 days.

The resulting hyperparameters of EMD and wavelets are provided in Table 3. One sees that EMD has different parameters for 5 s and 0.1 s windows—there is an upper bound in the IMFs in 2 of 3 cases of the 0.1 s window and no such bound is observed for the 5 s observation window experiments. Wavelets filter out the first scale for the 50 kHz sampling rate, 0.1 s window in 2 of 3 folds, as well as in all the folds of 1 kHz, 5 s window. The first wavelet scale corresponds to the highest frequencies out of the spectrum.

In the second stage of signal processing optimisation, we aim to find the optimal order of the applied methods. The results are provided in Table 4. In the optimised configuration, both EMD and wavelets are applied consistently across all the folds of the 0.1 s window experiments. The results are less consistent for the other two configurations: no pre-processing is done in 2 of 3 folds for 5 s window, 50 kHz sampling rate and in 1 fold of 1 kHz, 5 s window. Finally, the non-linear energy operator was not applied to any of the folds of any of the datasets.

#### Latency Tests

Below we report the evaluation times of the different components of the signal processing pipeline. Namely, we distinguish feature extraction and signal processing components. The evaluation times are reported in Table 5. Additionally, we observe that EMD takes the largest fraction of the evaluation time of up to 95%. While wavelets, when applied sequentially with the EMD, take less than 10% of the whole signal processing time.

### 3.3. Machine Learning Pipeline

When optimising the machine learning pipeline, we follow a uniform procedure for both estimators and all the experiment configurations. The feature space dimensionality obtained from the TSFEL package is 2016 input features and is consistent across all the experiments. Due to the large feature space, the feature selection is done in an annealing fashion, where the RFECV step decreases with the decrease of the feature space size. The feature number thresholds are: 700, 350, 125, 75, 37, 17, and 8, and the corresponding steps are: 400, 100, 50, 25, 12, 6, and 3. The final sizes of the feature space for both estimators, all the cross-validation folds, and experiments are provided in Table 6. Being a linear estimator, logistic regression on average is optimised to smaller feature spaces. Being an ensemble estimator, constructed of unstable base estimators (decision trees), CatBoost has more diversity in the feature space between folds.

The estimator hyperparameters are optimised after the feature selection. The particular optimised hyperparameter values are available in the published reproducibility package and not reported in the main body of the manuscript as largely influenced by the optimised feature spaces and can hardly be directly related to the results of the study. The classification performance for both estimators is provided in Table 7. At this point, there is no observable performance supremacy in any of the configurations. The worst performance is observed for the 50 kHz sampling rate, 0.1 s window, and the other two configurations perform similarly.

In order to investigate the classification patterns, we report a confusion matrix of the worst-performing experiment configuration—50 kHz sampling rate, 0.1 s time window, CatBoost classifier in Figure 2. Bearing failure entries are classified almost perfectly. The largest fraction of misclassified entries (>50%) is observed for the Horizontal Misalignment state. Moreover, Shaft Imbalance has 21% of entries misclassified. Interestingly, all Normal state entries are classified correctly; however, the Normal state is often confused with Shaft Imbalance and Horizontal Misalignment. The potential underlying reasons for such results are detailed in the Section 4 discussion.

#### Latency Tests

Below, we communicate the evaluation times of the utilised machine learning algorithms at the test phase. In Table 8, we report the time taken for a single entry classification for CatBoost and logistic regression. As one can see, CatBoost takes around one order of magnitude more time than the logistic regression classifier. While the CatBoost evaluation times largely depend on the number of input features, they are less pronounced for logistic regression where mean times of different configurations are within standard deviations of the measurements.

### 3.4. Statistical Evaluation

In order to better understand the results in terms of the experiment comparison as well as address the research questions, we perform the statistical tests and compute the effect sizes. We report these results in Table 9. In the tests, we expect the absolute errors to be statistically smaller in the alternative hypotheses. Median sample values are skewed with respect to the mean values, indicating a non-normal distribution of the data. This is especially evident for logistic regression, RQ1 experiment, where the alternative hypothesis mean error is less than that of the null hypothesis (medians have the opposite relationship); however, the *p*-value is still 1.0. This is due to the Wilcoxon test nature—it assesses the number of entries satisfying the condition, rather than the mean. The opposite situation with the medians indicates that even though by the mean absolute error the alternative hypothesis shows supremacy, it is not supported by the majority of entries, and the test confirms that.

Taking the Bonferroni corrections into account, we obtain the corrected significance level α=0.05/4=0.0125.

## 4. Discussion

In the current section, we reflect on the obtained results of the experiments. First, we address generally interesting points in the results, then, assess the significance of the results, limitations of the study, and implications for practitioners, and finally, discuss the future work.

In Table 3 one notices that EMD shows the best performance for the 5 s window with no constraints on IMFs. At the same time, constraints are introduced for the 0.1 s window. The potential reason is that spectral components which make sense for the longer window cannot be used for the shorter one and hence should be filtered out. This is also supported by the fact that EMD is known to perform well at filtering noisy non-linear non-stationary data [26]. We see a different picture in the optimisation results for wavelets transformation. Concretely, the lowest scale (highest frequency) is not filtered out for the 50 kHz sampling rate, 5 s window and filtered for the rest. This indicates that the highest frequencies are utilised by the model only in the setting of the high sampling rate and the larger time window. Nevertheless, in Table 4, one notices that both methods are used simultaneously only in 4 cases out of 9. Their sequence is always preserved and contradicts the sequence proposed in [2]. There might be multiple reasons for that, such as different data and experiment settings, and slightly different sets of methods. Interestingly, the non-linear energy operator led to a substantial increase of AICs and was excluded in all the experiment configurations, indicating that this method is not suitable for the considered dataset or the feature space. This result suggests that the sequence of the applied methods as well as the methods themselves are quite problem-specific and should be optimised on a per-configuration basis. This highlights the importance of the proposed method, allowing systematic, data-driven optimisation of the signal processing pipeline.

Considering the confusion matrix (Figure 2), there is an evident change in the classification performance between different failure modes. Namely, bearing failures are almost perfectly classified, even in the worst-performing configuration (the one reported). We hypothesise that having more rotating elements, bearing failures have a rich vibration spectrum which allows reliable separation of the bearing failures from the rest of the classes. Interestingly, we also do not see a performance drop in the 1 kHz sampling rate configuration associated with the worse bearing failures classification. At rotation frequencies of up to 60 Hz with 8 balls in the bearings, we would expect to see the first and second bearing harmonics within the 1 kHz frequency. In our classification, we distinguish the failures of overhang and underhang bearings with no further failure type classification. We hypothesise that two accelerometers mounted close to different bearings allow the failures to be distinguished based on the harmonic band intensity at the associated accelerometer. We highlight that, when optimising the sampling rate, it is essential to take into account the physical and mechanical properties of the system supported by domain knowledge.

Performance-wise, the proposed methodology is at least on a par with the existing works on the dataset [6,7,8]. If we neglect the different cross-validation strategies across studies, the accuracy obtained in the current work improves the state-of-the-art performance by above 1% in the absolute scale.

We highlight that the reported performances vary by up to 2% across papers. Considering the size of the dataset, these differences translate into a correct classification of around 40 entries. Considering the mentioned lack of a uniform framework for comparing the results across studies, the sources and significance of these differences are unclear. Since no evaluation times are reported in the reference papers, we withhold from comparing the latencies of the methods.

Latency-wise, the considered configurations show (Table 5 and Table 8) that the bottleneck in the processing speeds is the signal processing component. Moreover, with the increase of the sensor readings per observation window from 5 k to 250 k, there is a non-linear growth in the evaluation time of around 3 orders of magnitude. The non-linearity is caused mostly by the nature of the applied signal processing methods, EMD in particular. The weaker source of non-linearity is a varying set of the applied methods—EMD is present in all 3 configurations of fold 0, and wavelets are applied only in two of them. While processing of the 6 data channels takes on average 3.7 s for a 50 kHz sampling rate, 0.1 s observation window, we state that it is feasible to classify entries at sub-second latency by either using multiple cores or by optimising the number of data channels. The above observations indicate the importance of the sampling rate and the observation window optimisation, as they potentially lead to a rapid non-linear increase in the evaluation times, making the whole pipeline impractical for the majority of applications and likely useless for latency-critical applications. Consequently, answers to the stated research questions are of critical importance for the evolving field of IoT.

Assessing the financial costs of the proposed approach, we assume a price of around 5 cents per one CPU hour. Neglecting the costs of data storage and bandwidth, we end up with 1000 USD spent for the computational resources (for 20 k of CPU hours) to run all the experiments reported in the paper. It is worth noting that more than 90% of these resources are required to optimise the 50 kHz, 5 s observation window configuration. We consider these expenses totally acceptable for commercial companies deploying IoT infrastructures.

### 4.1. RQ1—Sampling Rate Impact

In the first research question, we study how the sampling rate affects the model performance (mean absolute error (MAE) of the output per-class probabilities) by comparing the 1 kHz and 50 kHz sampling rate experiment configurations. We report the *p*-values in Table 9. We reject the null hypothesis for the CatBoost estimator, but we cannot reject it for the logistic regression case. Hence, we claim that for the CatBoost classifier the higher sampling rate leads to a significantly better performing and more confident model for the considered machine operation frequency range. Being significant, effect sizes indicate that these results are likely to hold in the statistical population represented by different machines and configurations.

### 4.2. RQ2—Time Window Impact

In the second research question, we investigate the impact of the observation time window on the model performance. Based on the *p*-values in Table 9, we reject the null hypothesis. Hence, we positively answer the research question claiming that the 5 s time window leads to a significantly better classification performance than the 0.1 s window. Similarly to RQ1, the effect sizes are significant. However, the absolute classification performance obtained for the 0.1 s is still high. This result suggests that the use of shorter observation windows should be decided on a case-by-case basis. Of course, when considering industrial applications, there are external sources of vibrations, and the amount of labelled data is limited. No doubt, it leads to a generally worse classification performance, and both the sampling rate and the observation window length should be chosen with the context in mind.

### 4.3. Limitations

It is important to highlight that the proposed experiment design is one of the many ways of achieving the stated objectives.

There are limitations rooted in the design decisions. Concretely, we first perform the hyperparameter optimisation of the signal processing methods and then optimise the order. It might lead to a sub-optimal optimisation result since with this design decision, we implicitly assume that the optimal hyperparameters identified for a single method would hold for the sequence of methods as well. While it is computationally not feasible to perform the hyperparameter optimisation simultaneously with the method sequence optimisation, one should keep this limitation in mind.

Another limitation arises when assessing the performance of the models using different metrics. As one can see in Table 7, accuracy, precision, recall, and f1 scores of the CatBoost estimator are worse for the 50 kHz, 5 s window than for the 1 kHz, 5 s window, which does not agree with the MAE metric and the outcome of the statistical test in Table 9. The metrics in Table 7 consider the output confidence threshold only and hence reflect a different aspect of the model performance than MAE. Since MAE is calculated on the model output class probabilities, it computes how far on average the predicted probabilities are from the true labels, and consequently, might be giving a more complete picture than the other metrics. We use this metric in the statistical tests as it allows comparing the data in a per-entry fashion, leading to a higher statistical power of the results.

We perform the experiments on a dataset with labels indicating particular failure modes. The use of the findings is limited to condition monitoring and can hardly be used for predictive maintenance. Moreover, all the data in the dataset are obtained from a single machine with different failure modes introduced, limiting the generalisability of the findings to multiple machines. Nevertheless, being general, the proposed approach to the signal processing pipeline optimisation can be used in the predictive maintenance setting and multiple machines data.

We use a large set of features in the study. By doing so, we aim to improve the generalisability of the results. If one uses spectral features outside of the current feature space, there is a chance that the RQ findings will hold, but it would require additional verification. The same holds for the used machine learning models and the obtained configurations of the signal processing pipeline.

Finally, when optimising the signal processing pipeline, AIC is computed for the logistic regression model. Even though the signal processing optimisation is performed on training data, the performance comparison of the two estimators would be biased towards the logistic regression model. The bias is evident in the reported performance of estimators for the 50 kHz sampling rate and 0.1 s observation window. Such a difference between the estimators might suggest that the proposed method is rather estimator-specific.

### 4.4. Implications for Practitioners

Using the demonstrated approach, practitioners are able to optimise their systems, backing the complexity of the pipelines with the data. As we saw, the optimisation allows to effectively detect components that do not contribute to the performance improvement and hence improve the efficiency of the data processing stack making it more suitable for use at the Edge level without need for transferring large data volumes. The optimisation is computationally demanding but should be done only once per problem or a family of problems.

The answers to the research questions contribute to the development of lower-latency and more cost-effective failure classification systems. Namely, a decrease in the sampling rate would lead to a performance loss; however, on an absolute scale, the loss is small and might be acceptable in many settings. At the same time, a smaller observation time window leads to a more pronounced performance and model confidence drops (based on effect sizes); hence, it is advised to systematically optimise the observation window size on a case-by-case basis. From our findings, one might conclude that the use of non-linear estimators is not necessary; however, this is to be formally studied in an industrial setting, where external sources of noise are present. Finally, we hope that the methodology used in the study will encourage more reproducibility and comparability in the field.

### 4.5. Future Work

There are several pathways the current study can be extended into. As we previously mentioned, the obtained classification performance will likely decrease in the real-world setting. Hence, it would be interesting to extend the findings to industry data with the external vibration sources acting as noise. Considering the complexity of getting the ground truth for a practical number of entries in an industrial setting, the noise can be artificially generated and injected into the available data. This would allow addressing the generalisation of the findings to the real-world setting. Another research direction is finding the thresholds of the sampling frequency and the observation window length, below which the classification performance substantially drops.

## 5. Conclusions

In the current study, we proposed a way of optimising the signal processing approach which is further integrated into the machine learning pipeline. Moreover, we have answered the first research question (RQ1), by finding out that the increased sampling rate significantly improves the classification performance (for 1 of the 2 classifiers). Moreover, we have answered the second research question by discovering that the increase of the analysis time window improves the classification performance. The findings of both research questions are complemented by the insignificant effect sizes. It suggests that the improvements coming from the increased sampling rate and the observation window length are likely to generalise to the statistical population. However, the absolute performance drop is moderate, making the use of the lower sampling rates and shorter observation windows feasible for cost-effective low latency applications. Discussing the findings, we detailed the limitations of the experiment design and the findings. We conclude that even though lower sampling rates, as well as sub-second observation time windows, lead to a statistically significant classification performance decrease, it is not supported by the effect sizes, and their performance on an absolute scale is still very high. Hence, they can be used for the failure mode classification in lower budget and lower latency IoT settings. Finally, we ensure the reproducibility of the study by providing the code base for repeating and extending the reported experiments [27].

## Figures and Tables

**Figure 1 sensors-21-06678-f001:**
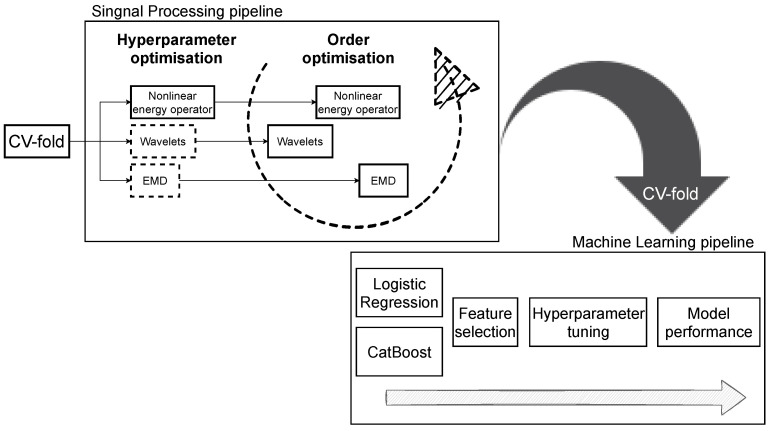
Diagram illustrating the optimisation steps of the signal processing and then machine learning (ML) pipelines. The dashed lines in the signal processing part represent elements optimised at the considered stage. The ML pipeline is run from left to right sequentially. After the optimal configuration for a signal processing pipeline is found, the dataset is processed and fed into the ML pipeline. At every step, the fixed set of features is extracted from the data and logistic regression is fitted and its goodness of fit is assessed (Akaike’s Information Criterion). The CV-fold corresponds to the cross-validation folds used throughout the study.

**Figure 2 sensors-21-06678-f002:**
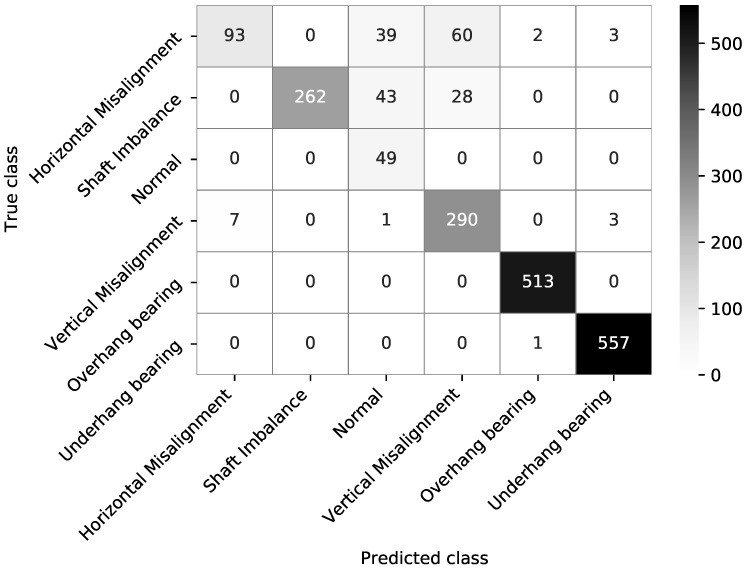
We present the confusion matrix of the worst-performing experiment—50 kHz sampling rate, 0.1 s time window, CatBoost classifier.

**Table 1 sensors-21-06678-t001:** The table provides details on the dataset, as it was used in the study.

Failure Mode	Number of Entries
Normal	49
Horizontal misalignment (shaft)	197
Vertical misalignment (shaft)	301
Shaft Imbalance	333
Overhang bearing	513
Underhang bearing	558
**Total**	1951

**Table 2 sensors-21-06678-t002:** The table provides details of the signal processing hyperparameter optimisation for EMD and wavelet methods. The parameters in the square brackets correspond to the sets of values used in the search space. The −1 value corresponds to the second largest available value and None indicates an absence of the bound. The ’shrunken’ space is used for the 250 k data points configuration and ’full’ for the 5 k points.

Method	Parameter Values	
	Shrunken	Full
EMD, IMF lower bound	[0, 1]	[0, 1, 2, 3, 4, 5]
EMD, IMF upper bound	[−1, None]	[6, 7, 8, 9, None]
Wavelet, lower scale	$2^{\left[0,1\right]}$	$2^{\left[0,1,2,3\right]}$
Wavelet, upper scale	$2^{\left[9\right]}$	$2^{\left[5,7,9\right]}$
Wavelet, type	[Morlet, Gaussian]	[Morlet, Gaussian]

**Table 3 sensors-21-06678-t003:** The table shows the final EMD and wavelet hyperparameters for all the experiments and cross-validation (CV) folds. The parameters are reported as “IMF lower bound, IMF upper bound” for EMD and “lower scale, upper scale, type” for wavelets.

CV Fold	EMD		
	50 kHz, 0.1 s	50 kHz, 5 s	1 kHz, 5 s
0	0, None	0, None	0, None
1	0, 8	0, None	0, None
2	0, 9	0, None	0, None
	**Wavelets**		
0	$2^{1}$, $2^{9}$, Morl	$2^{0}$, $2^{9}$, Morl	$2^{1}$, $2^{9}$, Morl
1	$2^{1}$,$2^{9}$, Morl	$2^{0}$, $2^{9}$, Morl	$2^{1}$, $2^{9}$, Morl
2	$2^{0}$, $2^{9}$, Morl	$2^{0}$, $2^{9}$, Gaus	$2^{1}$, $2^{9}$, Morl

**Table 4 sensors-21-06678-t004:** The table shows the final order of the signal processing methods applied for all the experiments and cross-validation (CV) folds. The sequences are reported in the same order they are applied. ’None’ corresponds to no signal processing performed.

CV Fold	50 kHz, 0.1 s	50 kHz, 5 s	1 kHz, 5 s
0	EMD, wavelets	EMD, wavelets	EMD
1	EMD, wavelets	None	None
2	EMD, wavelets	None	EMD

**Table 5 sensors-21-06678-t005:** The table shows the latencies of the signal processing pipelines for optimised configurations of all the experiments. We separately report the feature extraction and the signal processing times. The tests are performed on the dataset fold 0, as the case with potentially longest evaluation times. The times are reported in seconds as mean and standard deviation over 100 pipeline runs.

Component	50 kHz, 0.1 s	50 kHz, 5 s	1 kHz, 5 s
Signal Processing	3.7 ± 1.06	2570 ± 538	2.3 ± 0.46
Feature Extraction	0.34 ± 0.0176	22 ± 1.13	0.33 ± 0.0188

**Table 6 sensors-21-06678-t006:** In the current table, we report the final numbers of features after the feature selection process for the 3 cross-validation folds, CatBoost and logistic regression estimators and all the experiments. The experiments are encoded as “sampling frequency, time window length”.

Estimator	Experiment		
	50 kHz, 5 s	50 kHz, 0.1 s	1 kHz, 5 s
CatBoost	16, 268, 716	191, 2016, 666	241, 566, 164
Logistic Regression	125, 200, 150	75, 200, 122	200, 263, 350

**Table 7 sensors-21-06678-t007:** In the current table, we report the f1-score, precision, recall and accuracy (Acc.) of the estimators for all the experiments. F1-score, precision, recall are weighted by class.

Experiment	Logistic Regression				CatBoost			
	F1	Precision	Recall	Acc.	F1	Precision	Recall	Acc.
1 kHz, 5 s	0.99	0.99	0.99	0.99	1.00	1.00	1.00	1.00
50 kHz, 5 s	1.00	1.00	1.00	1.00	0.99	0.99	0.99	0.99
50 kHz, 0.1 s	0.94	0.97	0.93	0.93	0.91	0.94	0.90	0.90

**Table 8 sensors-21-06678-t008:** The table shows the latencies of the machine learning algorithms for the optimised feature spaces of all the experiments. The tests are performed on the dataset fold 0. The times are reported in seconds as mean and standard deviation over 100 runs.

Component	50 kHz, 0.1 s	50 kHz, 5 s	1 kHz, 5 s
CatBoost	$4.5\pm 0.029\times 10^{-3}$	$1.45\pm 0.057\times 10^{-3}$	$4.8\pm 0.019\times 10^{-3}$
Logistic Regression	$2.3\pm 0.64\times 10^{-4}$	$1.73\pm 0.16\times 10^{-4}$	$1.95\pm 0.27\times 10^{-4}$

**Table 9 sensors-21-06678-t009:** The table communicates the statistical test outcomes as well as supporting sample statistics. “Alt.” and “Null” correspond to the alternative and null hypothesis, respectively. Below those, we indicate the particular experiment representing the hypotheses. The tests and the sample statistics are computed on the absolute per-entry-per-class classification errors.

Statistics	Research Question			
	**CatBoost**			
	RQ1		RQ2	
One-tailed Wilcoxon test *p*-value	<0.001		<0.001	
Test Statistics	245,529.0		224,899.0	
Effect Size (Hedges *g${}_{av}$*)	0.165±0.057		0.64±0.060	
	**Test Groups**			
	Alt.	Null	Alt.	Null
	50 kHz, 5 s	1 kHz, 5 s	50 kHz, 5 s	50 kHz, 0.1 s
Mean (Absolute Error)	4.1$\;\times \;10^{-3}$	8.4$\;\times \;10^{-3}$	4.1$\;\times \;10^{-3}$	0.044
Median (Absolute Error)	3.5$\;\times \;10^{-4}$	1.36$\;\times \;10^{-3}$	3.5$\;\times \;10^{-4}$	1.22$\;\times \;10^{-3}$
Standard Deviation	0.025	0.027	0.025	0.085
	**Logistic Regression**			
	RQ1		RQ2	
One-tailed Wilcoxon test *p*-value	1.0		<0.001	
Test Statistics	1,050,300.0		362,824.0	
Effect Size (Hedges *g${}_{av}$*)	0.107±0.057		0.48±0.060	
	**Test Groups**			
	Alt.	Null	Alt.	Null
	50 kHz, 5 s	1 kHz, 5 s	50 kHz, 5 s	50 kHz, 0.1 s
Mean (Absolute Error)	1.17$\;\times \;10^{-3}$	3.3$\;\times \;10^{-3}$	1.17$\;\times \;10^{-3}$	0.028
Median (Absolute Error)	7.9$\;\times \;10^{-6}$	4.0$\;\times \;10^{-6}$	7.9$\;\times \;10^{-6}$	1.16$\;\times \;10^{-4}$
Standard Deviation	0.0123	0.026	0.0123	0.077

## Data Availability

The study reproducibility package is made public through Zenodo [27].

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
