# Peer review of "Cost-Effective Vibration Analysis through Data-Backed Pipeline Optimisation"

_sensors, 2021, doi:10.3390/s21196678_

Round 1

Reviewer 1 Report

The article presents an effective vibration analysis and fault diagnosis scheme for IoT systems. I read the article with much interest, and founded interesting. However, I have a comment that need to be addressed. Resonance band demodulation (also called envelope analysis) is a very effective technique for extracting bearing failure features. Generally, the resonance frequency band is higher than 1kHz. This situation also needs to be discussed.

Reviewer 2 Report

This paper introduces a Cost-effective vibration analysis through data-backed pipeline optimisation. The proposed optimisation approach, as well as statistically supported 15 findings of the study, allow for an efficient design of IoT vibration analysis systems, both in terms 16 of complexity and costs, bringing us one step closer to the widely accessible IoT/Edge-based 17 vibration analysis.

As above, the manuscript presents interesting and novel approach for IoT vibration analysis. It can be accepted after major revision.

  1. The manuscript presents the results of the proposed approach. However, the comparison with other machine learning or deep learning method is absent. The coresponding comparison should be presented to demonstrate the efficiency and performance.
  2. The statement in Introduction for the industrial problem for solving should be introduced more detailly, such as, the computing capacity required for actual operation in industrial scenarios.
  3. The actual calculation delay and the signal processing and inference speed of each method; although the paper does a complete hypothesis, inference and simulation verification, but lack of actual implementation results, may still result unpredictable (failure) in applications.

Round 2

Reviewer 2 Report

The manuscript can be accepted for publication.